# What Is the Clinical Evidence Supporting Trauma Team Training (TTT): A Systematic Review and Meta-Analysis

**DOI:** 10.3390/medicina55090551

**Published:** 2019-08-30

**Authors:** Michael Noonan, Alexander Olaussen, Joseph Mathew, Biswadev Mitra, De Villiers Smit, Mark Fitzgerald

**Affiliations:** 1National Trauma Research Institute, Melbourne 3004, Australia; 2Trauma Service, The Alfred Hospital, Melbourne 3004, Australia; 3Emergency & Trauma Centre, The Alfred Hospital, Melbourne 3004, Australia; 4Department of Community Emergency Health and Paramedic Practice (DCEHPP), Monash University, Melbourne 3199, Australia; 5Department of Epidemiology and Preventive Medicine, Monash University, Melbourne 3004, Australia

**Keywords:** Trauma Team Training, wounds and injuries, education, simulation, mortality

## Abstract

*Background and Objectives:* Major trauma centres manage severely injured patients using multi-disciplinary teams but the evidence-base that targeted Trauma Team Training (TTT) improves patients’ outcomes is unclear. This systematic review aimed to identify the association between the implementation of TTT programs and patient outcomes. *Materials and Methods*: We searched OVID Medline, PubMed and The Cochrane Library (CENTRAL) from the date of the database commencement until 10 of April 2019 for a combination of Medical Subject Headings (MeSH) terms and keywords relating to TTT and clinical outcomes. Reference lists of appraised studies were also screened for relevant articles. We extracted data on the study setting, type and details about the learners, as well as clinical outcomes of mortality and/or time to critical interventions. A meta-analysis of the association between TTT and mortality was conducted using a random effects model. *Results*: The search yielded 1136 unique records and abstracts, of which 18 full texts were reviewed. Nine studies met final inclusion, of which seven were included in a meta-analysis of the primary outcome. There were no randomised controlled trials. TTT was not associated with mortality (Pooled overall odds ratio (OR) 0.83; 95% Confidence Interval; 0.64–1.09). TTT was associated with improvements in time to operating theatre and time to first computerized tomography (CT) scanning. *Conclusions*: Despite few publications related to TTT, its introduction was associated with improvements in time to critical interventions. Whether such improvements can translate to improvements in patient outcomes remains unknown. Further research focusing on the translation of standardised trauma team reception “actions” into TTT is required to assess the association between TTT and patient outcome.

## 1. Introduction

Injury and trauma related deaths affect all age groups and have no geographical boundary. By 2030, the World Health Organisation (WHO) predicts that road traffic crashes alone (a small proportion of total trauma deaths) will rank fifth in global cause of death; only surpassed by cardiovascular and respiratory illness [1].

The introduction of trauma systems has had a major impact on trauma related deaths [2]; a key tenet of these systems being the transportation of major trauma patients to designated high volume “Trauma Centres”. It is estimated that centralised care in major traumas centres is associated with a 15% reduction in mortality for severely injured patients [3].

Since the introduction of large-scale trauma systems, hospital-based trauma reception and resuscitation has been delivered by multi-disciplinary teams. Although it is difficult to separate the impact of trauma team-delivered care from that of trauma systems as a whole, there is evidence that the introduction of trauma teams has significantly improved patient outcomes [4,5]. Teams make fewer mistakes than individuals [4], however bringing individual “experts” together to perform a specified task does not automatically ensure that they will function effectively as a team [6]. Effective trauma teamwork depends on the willingness of clinicians from diverse backgrounds to effectively communicate and collaborate to achieve shared goals. In addition, effective trauma teams must be self-reflective and open to learn from shared experiences; collectively, these are known as ‘non-technical skills’. It follows that ineffective trauma team performance cannot be attributed solely to inadequate knowledge or skills of the individual team members, but from deficits in the ‘non-technical skills’ of the team [7,8]. “Training” trauma teams to improve both technical and non-technical skills has therefore been strongly supported by most jurisdictions with trauma systems [8].

Despite the reported benefits of trauma teams, a standardised approach to team-based trauma reception and resuscitation has yet to be agreed upon. Advanced Trauma Life Support (ATLS) version 10 continues to focus on a serial approach to trauma care suited to the single provider [9] and the World Health Organisation (WHO) trauma checklist [10] provides a “safety” framework, however, it falls short of defining the workflow and activities of a high-functioning trauma team. Recent work by Fitzgerald et al. [11] attempted to define these activities in greater depth, however, further research is required to understand its impact.

As a result, education and training content and methods vary widely between providers of Trauma Team Training (TTT). Didactic methods have the benefit of teaching larger groups at a low cost but may be hindered by lack of participant engagement. Face-to-face sessions and high-fidelity simulation, on the other hand, may not be as readily feasible or affordable in certain settings, but are advantageous in terms being closer to reality and usually more engaging. Research into the efficacy of TTT has identified few suitable outcome measures and as such, the effect of TTT on patient outcomes is not well known [8]. Team-training in other critical care areas has been well received by staff and shown to positively change behaviours; however, correlation with clinical outcomes is limited [12]. This systematic review aimed to identify the evidence available linking the association between implementation of TTT programs and patient outcomes of mortality, morbidity, and time intervals to critical interventions in trauma resuscitation).

## 2. Materials and Methods

### 2.1. Search Strategy

This systematic review searched the literature without any language or time restrictions and reported according to the Preferred Reporting Items for Systematic Reviews and Meta-Analyses (PRISMA) guidelines. Our broad lay term question was “Does TTT improve patient outcomes?”. We used a combination of subject headings and keywords (Appendix A
Table A1).

### 2.2. Eligibility Criteria

We included studies researching patients (adults and paediatrics) who suffered trauma in any setting (both metropolitan and rural, as well as developed and developing countries) and any severity (e.g., Glasgow Coma Score, Injury Severity Score). The intervention or exposure had to be directed towards staff caring for the trauma patients and who had undertaken any form of TTT. We compared with either no TTT or a period before TTT as comparison (depending on study design). The primary outcome was the mortality incidence (at a time point defined by the researchers, hospital discharge or at 28-days). Secondary outcomes were morbidity and time intervals to critical interventions in trauma resuscitation (e.g., time to operating theatre, intubation, CT scanning, transfer), as these markers correlate with morbidity and mortality. Studies had to include at least one of those outcomes to be eligible. We included randomised controlled trials and controlled trials, as well as observational cohort and case control studies.

Articles were excluded if they were deemed to be in the (i) wrong setting (e.g., not trauma, [13], (ii) no intervention just observation [14], (iii) wrong outcome measure (e.g., simulation [15], checklists [16], or self-rated confidence [17]). We also excluded article type (i.e., case reports and case series, as well as editorials, letters and conference abstracts) due to the high risk of bias.

### 2.3. Information Sources

We searched for articles from four databases (OVID Medline, PubMed, and The Cochrane Library (Cochrane Database of Systematic Reviews, Cochrane Central Register of Controlled Trials (CENTRAL)), extending from the databases’ commencement to 10 April 2019. We also screened reference lists of all selected studies for relevant articles that might not have been captured by the search strategy listed above. Protocol for the review was published on PROSPERO, the International prospective register for systematic reviews (PROSPERO CODE CRD42019131179).

### 2.4. Study Selection

Following the search, duplicates were removed using EndNote X7 (Clarivate Analytics, Philadelphia, PA, United States) and titles were independently screened by two authors (M.N. and A.O.). The abstracts of the identified studies were subsequently appraised for eligibility independently by the same two authors. The resulting studies then underwent full-text review to determine appropriateness of inclusion in the qualitative synthesis phase. Consensus resolved any disagreements concerning inclusion decisions.

### 2.5. Data Extraction and Analysis

From the included papers, we extracted data on study setting and size, methodology, participants, the intervention and the educational details (including length of the course and method) and the studies outcome measurements related to patient outcomes. We supplemented the pre-hoc defined outcome measures by an iterative approach based on findings from the individual studies.

### 2.6. Definitions

In terms of patient outcomes, we primarily focused on mortality. Morbidity was a secondary outcome. Thirdly, time to diagnosis (e.g., time to CT) and time to treatment (e.g., time to Operating Theatre (OT) or Endotracheal tube (ETT)) were included as they may function as valid surrogates for patient outcomes [18]. Both these outcomes would be regarded as Kirkpatrick levels 4—the highest level (i.e., 4a meaning a change in organisational practice, and level 4b meaning benefits to the patients) [19].

### 2.7. Assessment of Quality of Identified Papers

Methodological quality of observational studies were performed using the Newcastle-Ottawa quality assessment scale (NOS) [20]. A score of 5 or below was considered low quality; a score of 6 or 7 was considered medium quality; a score of 8 or 9 was considered high quality. Two independent authors (M.N. and A.O.) assessed the risk of bias and reached the same score without needing to go to consensus.

### 2.8. Meta-Analysis

Heterogeneity between studies was assessed using the I^2^ statistic. We conducted the meta-analysis using Stata version 13.1 (StataCorp, College Station, TX, USA), using the metan [21] command, reporting OR and using the DerSimonian & Laird random effects model. We used the random effects model in case of significant heterogeneity, as it is the generally accepted practice despite not being a complete solution to heterogeneity [22].

## 3. Results

### 3.1. Study Selection

A total of eleven hundred and thirty-six records were identified through database searching. After removal of duplicates, 1029 titles were available for screening. From this, 18 abstracts were reviewed. There was near perfect inter-rater reliability (Agreement 99.2%, kappa 0.71, *p* < 0.01). Following abstract screening, 18 unique full text studies were identified for review, of which nine were finally included, and seven in the meta-analysis on mortality (Figure 1).

### 3.2. Study Characteristics

Nine studies [7,23,24,25,26,27,28,29,30] comprising 5683 patients, were included in the final qualitative synthesis (Table 1). Studies were published between 2010 and 2018. The individual study size ranged from 144 to 2389 patients. The studies were conducted predominantly in the USA (*n* = 6), with the remaining three studies in Australia, Rwanda and China. All the studies were observational in nature using a before-after intervention design.

### 3.3. Results of Individual Studies

The studies employed various training interventions ranging from simulation to didactic teaching. The training spanned from hours to days. The follow up period ranged from 3 months to 4 years (Table 2).

### 3.4. Risk of Bias

The individual studies generally had a low risk of bias, with 6 out of 10 studies scoring high (8 or 9) on the Newcastle-Ottawa scale. (Table 3).

### 3.5. Risk of Bias Within Studies

The risk of bias within the individual studies was generally low (Table 3).

### 3.6. Mortality

Seven studies reported on mortality (two of which measured mortality at hospital discharge and 5 which were unspecified in the time frame) [7,23,24,25,27,28,29]. None of the studies found a statistically significant change in mortality (Figure 1). The studies were relatively homogenous, with 29.2% of the variation in OR attributable to heterogeneity (I^2^ = 0.292, *p* = 0.205) (Figure 2). Pooled overall OR was 0.83 (95% CI 0.64–1.09). The event rate in the pre-TTT group was 263/2477 (10.6%) compared to 316/3268 (9.7%) in the post-TTT group. Crude combined counts yielded a relative risk reduction of 8.9%.

### 3.7. Mortality-Subgroup Analysis

Petroze et al. [28] found that among patients with severe head injury (Glasgow Coma Scale (GCS) 3–5), the mortality was decreased from 84% in the pre-education group to 63% in the post-intervention group (*p* = 0.04, OR 0.32 (95% CI 0.10–0.99). Even with a more conventional grouping of severe head injury (i.e., GCS 3–8), the mortality was still significantly reduced from 59%–37% (*p* = 0.009, OR 0.42 (95 % CI 0.22–0.81).

### 3.8. Secondary Outcomes

None of the studies reported on morbidity outcomes. There was a general reduction in time to critical interventions in the resuscitation (Table 4). Two studies examined time to operating theatre and two studies examined the time to CT, both finding both clinically and statistically significant reductions. There was no statistically significant difference in time to ETT. ED length of stay was studied by 6 studies, 2 found no difference, 3 found a reduction and 1 found a significant increase from 4.88 (Inter Quartile Range (IQR) 2.03–8.05) h before to 7.17 (IQR 2.88–14.17) h post, (*p* < 0.001). There was no difference in ICU or hospital LOS in the two studies that examined that. Three studies assessed and found a statistically significant reduction in the ED transfer time to receiving trauma centres.

## 4. Discussion

This systematic review found a small number of manuscripts that had reported on the effect of trauma team training. No statistically significant reduction in mortality following TTT was demonstrated; however, key outcome measures, like time to OT and CT, were shown to improve.

TTT may have a greater benefit in groups with higher injury severity. While Capella et al. did not find a statistically significant reduction in mortality post TTT, their demonstrated mortality reduction (13.1% to 8.5%; *p* = 0.121) may have reached significance if their pre- and post-TTT groups had higher Injury Severity Score (ISS) values than they observed (ISS 11.6 vs. 14.0, *p* = 0.036). This follows the work of Cohen et al. [31], which showed an 8% reduction in mortality in a severely injured cohort (ISS ≥ 25) following the introduction of team-based trauma care.

All studies the that examined time to ‘critical interventions’ in trauma resuscitation found that time to OT and time to CT were improved following TTT. In addition, Hong et al. showed an improvement in time to establishing a definitive airway following TTT. This reduction in time to interventions was paralleled in the population of obstetrics and gynaecology patients (33.3 min versus 21.2 min, *p* = 0.03) following team training [13]. We believe that any reduction in time to critical interventions in severe trauma should be considered significant, as it provides the opportunity for marginal gains in outcomes.

Self-related scores such as ‘confidence’ and ‘competence’ are often used to assess the efficacy of TTT. We did not include studies that exclusively reported these measures, however, some included studies contained them. Although recent evidence suggests that self-assessment scores of “competence” rank higher than the rating of experts, [32] attention to these ‘softer’ outcomes remains a topic of interest in medical education, particularly in the assessment of team-based non-technical skills. To date, it is not clear whether enhancing staff confidence and competence translates to meaningful patient outcomes.

Educational delivery was generally succinct, with TTT being delivered in one day. No studies reported “blended” online and face-to-face content delivery methods, however, this may reflect the “era” in which the studies were undertaken rather than their educational validity. Although such methods may be criticised for reducing learner engagement, they may also assist in educational efficiency for faculty and in return, improve dissemination of training across a trauma system. The optimal content and andragogy approach to TTT was beyond the scope of this systematic review, however, recognising the variety of content delivery methods utilised by these studies, we suggest that a mix of blended content, didactic face-to-face training and high-fidelity simulation is both achievable and effective and should be considered as part of any TTT program.

Regarding the content (technical vs. non-technical) of TTT programs, Kappel et al. [26] tried to distinguish if it was team “communication” or the Rural Trauma Team Development Course (RTTDC), (a training/educational program developed by the American College of Surgeons) or a combination of both that impacted their primary outcome. All three groups showed a significant reduction in “decision time to transfer” trauma patients when compared to those with no training. Pragmatically, the TTT courses contained in this systematic review all contained elements of technical and non-technical training. The best “mix” of these elements is unknown but will be likely learner- and system-dependent.

Important limitations of this systematic review and of the included studies must be acknowledged. Firstly, TTT was poorly defined and the outcomes of successful adoption diverse. The included studies were all observational studies. The absence of RCTs brings concerns of inherent biases to the surface. In addition, few studies could be meta-analysed with respect to our primary outcome; this primarily relates to a lack of data examining the effects of TTT on mortality. Mortality was also infrequently defined by a time frame, but presumably related to hospital discharge or earlier as no papers described a patient follow up process. The effect of TTT on other outcomes, like improvements in teamwork scores (i.e., T-NOTECHS), efficiency of task completion has been demonstrated, [29] however, we did not include studies that did not report clinical patient outcomes. Finally, significant heterogeneity exists amongst the included studies in terms of the TTT, the participants and the settings. More broadly, this systematic review highlights the challenges faced by researchers studying TTT. TTT is not only heterogenous, more importantly standardised trauma team reception “actions” (i.e., clinical steps), the basis of any TTT are only recently being defined [11]. The value of any educational intervention is limited by the quality and rigor of the content being taught. Perhaps the first step in understanding the true impact of TTT is in the broad adoption of standardised trauma team reception “actions”, which can serve as a benchmark against which training can be measured.

## 5. Conclusions

The adoption of TTT has not been prospectively measured in respect to mortality outcomes. Of the few studies published, most are observational narratives of TTT and the uptake of new skills and behaviours amongst the TTT participants. This small set of non-randomised observational studies failed to demonstrate mortality benefit. Important clinical surrogates, such as time to OT and CT may improve after TTT. Further research should focus on the translation of standardised trauma team reception “actions” into TTT so that the impact of team-based training of these actions can be studied.

## Figures and Tables

**Figure 1 medicina-55-00551-f001:**
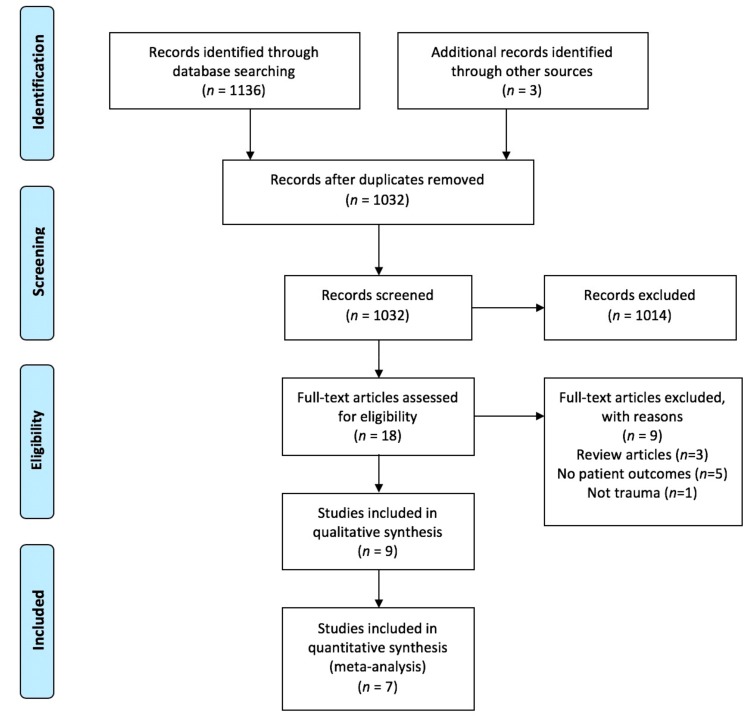
Study selection.

**Figure 2 medicina-55-00551-f002:**
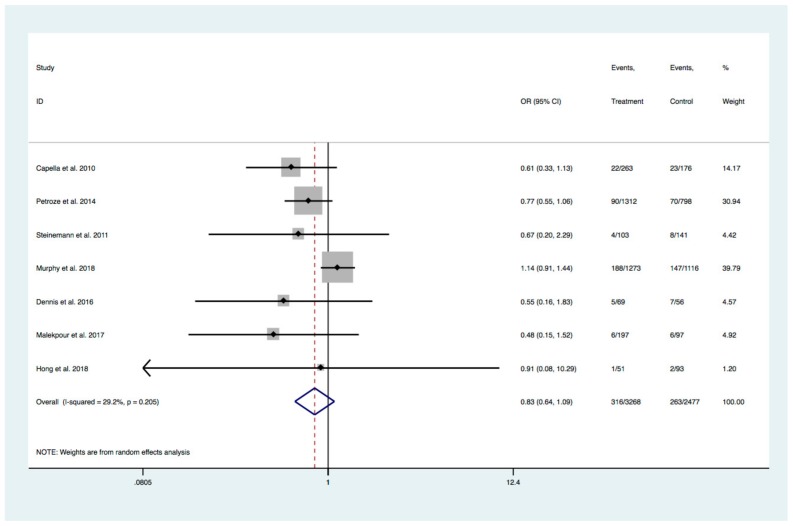
Effect of TTT on Mortality.

**Table 1 medicina-55-00551-t001:** Study characteristics.

Study	Country	Setting	Study Type	Trauma Centre?	Number of Learners	Learners
Capella et al. 2010 [23]	USA	Level I Trauma Centre	Prospective Pre/post test Interventional Uncontrolled Unblinded	Yes	114	> 80% of TT = 114 Surgery residents (*n* = 28) Faculty surgeons (*n* = 6) ED nurses (*n* = 80)
Dennis et al. 2016 [24]	USA	Rural non-trauma referring hospitals	Prospective Pre/post test Interventional Controlled Unblinded	No	Staff at 6 rural, non-trauma hospitals	Participants ED physicians, ED midlevel providers, ED nurses, respiratory therapists, radiology technologists, laboratory technologists, and prehospital personnel.
Hong et al. 2018 [25]	China	University-affiliated hospital	Retrospective review with propensity matching, before and after quality improvement prospectively implemented	NA	NA	6 ED nurses and doctors
Kappel et al. 2011 [26]	USA	Rural level III and level IV trauma centres USA	Multi Institutional 3-month longitudinal study Pre/Post Interventional Uncontrolled Unblinded	Yes	18 hospitals	Medical personnel located throughout the state of West Virginia representing 21 of a possible 32 designated trauma facilities
Malekpour et al. 2017 [27]	USA	3 non-trauma referring facilities	Retrospective Pre/post test Observational Uncontrolled Unblinded	No	NA	NA
Murphy et al. 2018 [7]	Australia	Level 1 Adult Trauma Centre	Retrospective Pre/post test Observational Uncontrolled Unblinded	Yes	324 participants in total, quarterly over 4 years, in groups of 20 per session	10 doctors (emergency, intensive care, anaesthetics, general surgery and trauma) 9 nurses and 1 radiographer.
Petroze et al. 2014 [28]	Rwanda	The Centre Hospitalier Universitaire Kigali (CHUK) 520-bed hospital in the capital of Kigali	Prospective Pre/post test Interventional Uncontrolled Unblinded	Yes	64	Faculty surgeons (*n* = 24) Trauma nurse auditors (*n* = 15) Faculty, residents and nurses (*n* = 25)
Steinemann et al. 2011 [29]	USA, Hawaii	Designated Trauma Centre (Level II) serving Hawaii and the Pacific Basin	Prospective Pre/post test Interventional Uncontrolled Unblinded	Yes	137	137 multidisciplinary trauma team members, including residents, ED and trauma attending physicians, nurses, respiratory therapists, and ED technicians
Zhu et al. 2016 [30]	USA	Level 2 Trauma Centre	Retrospective Pre/post test Observational Uncontrolled Unblinded	Yes	163	101 nurses, 8 physicians, 33 pre-hospital personnel, 13 respiratory therapists, 8 radiology technicians. excluding one lab tech + 5 unknown participants

Abbreviations: NA = Not available; ED = Emergency Department; TT = Trauma Team.

**Table 2 medicina-55-00551-t002:** Details the different intervention used by included studies.

Study	Number of Patients/Cases-TOTAL	Number of Patients/Cases-PRE Intervention or Control	Number of Patients/Cases-POST Intervention or Experimental Group	Intervention	Didactic	Simulation	Use of Feedback	Duration	Time of Evaluation
Capella et al. 2010 [23]	439	176	263	TeamSTEPPS (focusing on Briefing, STEP, CUS, Call Outs, and Check Backs) + simulation	2 h	Yes—(2 h) 7 Multidisciplinary sessions with 2 senior and 2 junior residents for each, as well as 1 attending and 2 to 3 ED nurses.	Video-taped, reviewed immediately by the entire team.	4 h	Pre and 3 months post
Dennis et al. 2016 * [24]	125	56	69	RTTDC	NA	NA	NA	NA	12 months pre and post
Hong et al. 2018 [25]	144	93	51	Locally developed TTT	Yes	Yes	No	4 days	Over 15 months
Kappel et al. 2011 [26]	308	272	36	RTTDC	NA	NA	NA	NA	Pre and 3 months post
Malekpour et al. 2017 [27]	276	97	179	RTTDC	NA	NA	NA	NA	2 years pre and post
Murphy et al. 2018 [7]	2389	1116	1273	Training program	3 × 60 min sessions	High-fidelity	Yes	8 h	4 years pre and post
Petroze et al. 2014 [28]	1373	798	575	Trauma education course	NA	NA	NA	6 days	6 months pre and post
Steinemann et al. 2011 [29]	244	141	103	The intervention was a multidisciplinary, HPS-based, in situ team training curriculum.	Yes	Yes	Yes	4 h	6.5 months pre and post
Zhu et al. 2016 [30]	257	114	143	RTTDC	NA	NA	NA	8 h + self-study	12 months pre and post

NA = Not available; RTTDC = Rural Trauma Team Development Course; * For the Dennis et al. study we only included the post-era data to allow the least biased interpretation.

**Table 3 medicina-55-00551-t003:** Risk of bias assessment.

Study	Selection	Comparability	Outcome	Total
Capella et al. 2010 [23]	4	0	3	7
Dennis et al. 2016 [24]	3	2	3	8
Hong et al. 2018 [25]	4	2	3	9
Kappel et al. 2011 [26]	3	0	2	5
Malekpour et al. 2017 [27]	4	1	3	8
Murphy et al. 2018 [7]	4	1	3	8
Petroze et al. 2014 [28]	4	1	3	8
Steinemann et al. 2011 [29]	4	0	3	7
Zhu et al. 2016 [30]	4	2	3	9

**Table 4 medicina-55-00551-t004:** Time to critical interventions in trauma resuscitation.

Study	Mortality	Time to OT	Time to CT	Time to ETT	ED LOS	ICU LOS	Hospital LOS	ED Transfer time
Capella et al. 2010 [23]	Pre 13.1% (of 176)Post 8.5% (of 263)*p* = 0.121	Pre 130.1 (82.7) minPost 94.5 (63.8) min*p* = 0.021	Pre 26.4 (14.5) versusPost 22.1 (11.7)*p* = 0.005	Pre 10.1 (6.8) versusPost 6.6 (4.2)*p* = 0.49	Pre mean 186.1 (151.0) minPost mean 187.4 (159.3) min*p* = 0.93	Pre 5.5 (6.4)Post 6.3 (6.8)*p* = 0.445	Post 7.6 (14)Pre 6.3 (5.8)*p* = 0.21	
Dennis et al. 2016 [24]	Pre 7/56Post 5/69				Pre 195 (120–251) minPost 122 (91–176) min*p* = 0.002			Pre 137.7 minPost 100.1 minDiff-in-diff = −41.241*p* = 0.03
Hong et al. 2018 [25]	Pre *n* = 2 (6.5%) toPost *n* = 1 (5.9%)*p* = na		Pre 47 (35.5–77) min versusPost 29.5 (18.5–36.5)*p* = 0.01	Airway establishmentPre 15 (3–28.25) minPost 40 (25.25–65.25) min*p* = 0.24	Pre 3 (1.62–5.75) hPost 2 (1–3)*p* = 0.05			
Kappel et al. 2011 [26]	NA							Statistically significant shorter from arrival to decision and from decision to transfer*p* < 0.05
Malekpour et al. 2017 [27]	Pre 6/97Post 6/197*p* = 0.354				Pre 4 (1–6)Post 4 (2–7)*p* = 0.295*Presumably hours			Pre 257.4 (110.8) min vs.Post 219.2 (86.5) min*p* = 0.002
Murphy et al. 2018 [7]	Pre 13.2% (n = 147/1116)Post 14.7% (*n* = 188/1273)*p* = 0.29	Pre 2.63 hrs (IQR 1.23–5.12) toPost 0.55 hrs (IQR 0.22–1.27)*p* < 0.001			Increased.Pre 4.88 h (IQR 2.03–8.05)Post 7.17 h (IQR 2.88–14.17)*p* < 0.001			
Petroze et al. 2014 [28]	Pre 8.8% (n = 70/798)Post 6.3% (*n* = 90/1312)*p* = 0.09							
Steinemann et al. 2011 [29]	Pre n = 8/141Post *n* = 4/103*p* = NS					Pre 1.9 daysPost 0.3 days*p* = NS	Pre 5.1 daysPost 3.4 days*p* = NS	
Zhu et al. 2016 [30]	NA				Reduction of 43 min (95% CI −72 to −14, *p* = 0.004)

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
