# Peer review of "What Is the Clinical Evidence Supporting Trauma Team Training (TTT): A Systematic Review and Meta-Analysis"

_medicina, 2019, doi:10.3390/medicina55090551_

Round 1

Reviewer 1 Report

This manuscript is a meta-analysis of Trauma Team Training looking its impact on clinical process and outcome measures.  As such, it is a worthwhile contribution to the literature.  Some revisions/clarifications are required in order to improve it. First, a paragraph in the introduction addressing the types of training available and their advantages and disadvantages would be useful.  Also, a comment on why time to CT and OR and ET are critical process markers in trauma care (i.e., correlation to morbidity and mortality) would be helpful.  In Table 1, the Murphy et al line lists 20/session but only 20 are listed as Learners (was there only 1 session).  Listing the number of sessions per training might clarify this.  A summary statement related to how many studies were good vs. moderate vs. poor in the text would be useful.  In the discussion, the term team reception "actions" is used several times.  Please expand on what is meant by it. 

Author Response

Cover letter

Thank you for review, comments and suggestions. These have been incorporated as indicated below, item by item and as seen in the tracked changes version. We remain open to any further suggestions to optimise the manuscript further.

Reviewer 1

This manuscript is a meta-analysis of Trauma Team Training looking its impact on clinical process and outcome measures.  As such, it is a worthwhile contribution to the literature.  Some revisions/clarifications are required in order to improve it.

First, a paragraph in the introduction addressing the types of training available and their advantages and disadvantages would be useful. 

Thank you - added

Didactic methods have the benefit of teaching larger groups at a low cost, but may be hindered by lack of participant engagement. Face-to-face sessions and high-fidelity simulation on the other hand may not be as readily feasible or affordable in certain settings, but are advantageous in terms being closer to reality and usually more engaging.

Also, a comment on why time to CT and OR and ET are critical process markers in trauma care (i.e., correlation to morbidity and mortality) would be helpful. 

Thank you - added

In Table 1, the Murphy et al line lists 20/session but only 20 are listed as Learners (was there only 1 session).  Listing the number of sessions per training might clarify this. 

Corrected

There were four programs each year with a total of 324 multidisciplinary trauma team staff at the study site completing the program during the 4 years post intervention period.

A summary statement related to how many studies were good vs. moderate vs. poor in the text would be useful. 

Added

In the discussion, the term team reception "actions" is used several times.  Please expand on what is meant by it. 

This means the clinical steps. Added

Reviewer 2 Report

Thank you for the opportunity of reviewing this systematic review and meta-analysis. This is a very well planned and well executed answer to a specific research question. The research method is well stated, the inclusion and exclusion criteria are well explained and well chosen, and the data analyses and presentations are straight-forward and well explained, as well as the limitations to the study.

There is one major concern:

One of the included study, Dennis et al 2016, had 123 pre-intervention (control) cases and 130 post-intervention (experimental) cases as shown in Table 2. However, a meta-analysis of the association between TTT and mortality as shown in the Figure 2 (the Forest plot) revealed only 61 in control and 69 in the experimental, and the mortality OR in this study revealed 4.69 (95% CI: 0.53-41.29). I reviewed the original article (reference 24), and found that there were no differences in mortality between the two groups, as concluded by the authors themselves. I think there might be some misunderstanding right here in this specific study (Dennis et al, 2016). In my view, the mortality ratio in pre-TTT group (control) should be 13/123 and the ratio in the intervention group should be 6/130. In this regards, the pooled OR of this meta-analysis study will be 0.78 (95% CI: 0.58-1.06) by a random effects model, still showing no significant difference in mortality between groups.

Author Response

Cover letter

Thank you for review, comments and suggestions. These have been incorporated as indicated below, item by item and as seen in the tracked changes version. We remain open to any further suggestions to optimise the manuscript further.

Reviewer 2

Thank you for the opportunity of reviewing this systematic review and meta-analysis. This is a very well planned and well executed answer to a specific research question. The research method is well stated, the inclusion and exclusion criteria are well explained and well chosen, and the data analyses and presentations are straight-forward and well explained, as well as the limitations to the study.

There is one major concern:

One of the included study, Dennis et al 2016, had 123 pre-intervention (control) cases and 130 post-intervention (experimental) cases as shown in Table 2. However, a meta-analysis of the association between TTT and mortality as shown in the Figure 2 (the Forest plot) revealed only 61 in control and 69 in the experimental, and the mortality OR in this study revealed 4.69 (95% CI: 0.53-41.29). I reviewed the original article (reference 24), and found that there were no differences in mortality between the two groups, as concluded by the authors themselves. I think there might be some misunderstanding right here in this specific study (Dennis et al, 2016). In my view, the mortality ratio in pre-TTT group (control) should be 13/123 and the ratio in the intervention group should be 6/130. In this regards, the pooled OR of this meta-analysis study will be 0.78 (95% CI: 0.58-1.06) by a random effects model, still showing no significant difference in mortality between groups.

This study looked at “The effect of RTTDC training on transfers from nontrauma centers to definitive care”. A pre/post analysis of trauma patients who were transferred from rural, nontrauma hospitals from 2012 to 2014. Patients from six rural hospitals that participated in an RTTDC course were compared with a control group of similar centers that did not participate in the course. Whilst there was 253 patients available for study (RTTDC group, n = 130; control group, n = 123); In the RTTDC group, 61 were in the “pre-era” and 69 “post-era”. In the control group; 67 were in the pre-era and 56 in the post-era. As per the paper’s table 2, there were 5 deaths from a total of 69 in the RTTDC post-era group and 7 deaths from the 56 in the control post-era group. We have interpreted the results as the “pre-era” cohort being for assessment of selection bias to show that the 6 hospitals that eventually got the TTT did not differ from 6 other hospitals. As such we only included this cohort in the analysis to take the least biased interpretation of the results. We have now added this description to the manuscript and updated the meta analysis.

Reviewer 3 Report

This systematic review is well written with appropriate methodology and good discussion. I have no major concerns other than grammatical errors.

・“Was” is redundant in line 81.

・You should add “reviews” to “The International resister of systematic reviews” in line 95.

・“Both” is duplicated in line 110.

Author Response

Cover letter

Thank you for review, comments and suggestions. These have been incorporated as indicated below, item by item and as seen in the tracked changes version. We remain open to any further suggestions to optimise the manuscript further.

Reviewer 3

This systematic review is well written with appropriate methodology and good discussion. I have no major concerns other than grammatical errors.

Thank you

“Was” is redundant in line 81.

Addressed

You should add “reviews” to “The International resister of systematic reviews” in line 95.

Added

 “Both” is duplicated in line 110.

Removed